

# Snowfall versus sub-shelf melt: response of an idealized 3D ice-sheet-shelf system to mass redistribution

Johannes Feldmann[1], Ronja Reese[1,2], Ricarda Winkelmann[1,2], and Anders Levermann[1,2,3]

[1]Potsdam Institute for Climate Impact Research (PIK), Potsdam, Germany
[2]Institute of Physics, University of Potsdam, Potsdam, Germany
[3]LDEO, Columbia University, New York, USA

*Correspondence to:* Johannes Feldmann (johannes.feldmann@pik-potsdam.de)

**Abstract.** Surface accumulation and sub-ice-shelf melting are key drivers for the flow dynamics of the Antarctic Ice Sheet and are most likely to change under future warming which leads to 1) higher snowfall and 2) stronger melting below ice shelves. Here we carry out conceptual simulations in which an equilibrium ice-sheet-shelf system is perturbed such that the increased sub-shelf melting is compensated by enhanced snowfall. Although the net surface mass balance of the whole system remains unchanged, the redistribution of mass leads to a dynamic response of the ice sheet due to changes in ice thickness, surface slope, ice-shelf backstress and ice discharge. In particular, we show that such forcing can lead to the counter-intuitive situation of a retreating ice sheet which gains mass, thus having a negative sea-level contribution but smaller ice-sheet extent. The ice-sheet evolution and the corresponding steady states are investigated varying relevant parameters that affect ice properties and bed geometry as well as for different magnitudes of mass redistribution. Furthermore, the ice-sheet response is analyzed with respect to the pattern of applied melting, i.e., the area over which melting is distributed and the location where it is applied. We find throughout the ensemble of simulations that after two decades, melting at the lateral ice-shelf margins induces more ice-shelf thinning, resulting in stronger grounding line retreat and transient ice discharge compared to melting adjacent to the central grounding-line section. Analyzing changes in ice-shelf backstress with respect to changes in the ice-shelf length and mean thickness, respectively, we show that a thickness change has up to four times more influence on the backstress of the ice shelf than a length change.

## 1 Introduction

The mass balance of the Antarctic Ice Sheet is dominated by accumulation of snow at its surface and discharge of ice into the ocean at its margins (Rignot et al., 2011; Shepherd et al., 2012). The main driver of the currently observed mass loss of the ice sheet is basal melting of its fringing ice shelves (Jenkins et al., 2010; Pritchard et al., 2012; Rignot et al., 2013; Paolo et al., 2015) which regulate the ice discharge from the inland ice sheet (Dupont and Alley, 2005; Favier et al., 2012; Asay-Davis et al., 2016; Reese et al., 2018). Observed consequences include speed-up, thinning and retreat of the grounded ice sheet (Shepherd et al., 2002; Jenkins et al., 2010; Joughin and Alley, 2011; Rignot et al., 2014; Konrad et al., 2018).

Future atmospheric warming will likely increase both the snowfall over the Antarctic Ice Sheet (Monaghan et al., 2008; Ligtenberg et al., 2013; Frieler et al., 2015a) and the oceanic heat content available for sub-ice-shelf melting (Rignot, 2002;





Hellmer et al., 2012; Spence et al., 2014; Schmidtko et al., 2014). An increase in surface accumulation leads to a higher rate of mass gain over the continent (Thomas et al., 2017). At the same time it can enhance ice loss due to increased discharge across the grounding line (Winkelmann et al., 2012). However, whether increased snowfall can balance accelerated ice discharge remains little studied (Frieler et al., 2015b).

Increased melting may lead to ice discharge and thus contribute positively to future sea level rise (e.g., Bindschadler et al., 2013; Levermann et al., 2014; Bamber and Aspinall, 2013; Joughin et al., 2014; Favier et al., 2014; Mengel and Levermann, 2014; Pollard et al., 2015; Bakker et al., 2017; Jackson et al., 2018). Melting of laterally confined ice shelves reduces the backstress that they exert on the upstream grounded ice (Thomas, 1973; Hughes, 1973; Pritchard et al., 2012; Wouters et al., 2015). As has been shown in idealized experiments such backstress reduction can be induced by both a decrease in the overall

ice-shelf thickness or in the length of the ice-shelf confinement (Dupont and Alley, 2006; Gagliardini et al., 2010). Furthermore, backstress is lower for wider ice-shelf confinements (Dupont and Alley, 2006; Goldberg et al., 2009; Gudmundsson et al., 2012). Gagliardini et al. (2010) found in conceptual simulations that the grounding-line position and the volume of an ice sheet are sensitive to changes in the degree of concentration of the melting to the grounding line even if the average melt magnitude remains the same. This suggests that not only the magnitude but also the location and distribution of ice-shelf thinning have

strong influence on the backstress and the corresponding ice-sheet response (Fürst et al., 2016; Reese et al., 2018). Usually, the highest basal melt rates of Antarctic ice shelves are observed close to the grounding line where the ice shelf is thickest which is typically downstream of channelized ice streams (Dutrieux et al., 2013). However, ice shelf melting as well as thinning patterns can be spatially very heterogeneous (Dutrieux et al., 2013; Paolo et al., 2015), including the possibility of comparatively strong melting in the more stagnant lateral parts of an ice shelf (Berger et al., 2017).

Here we carry out ensemble simulations of an idealized, three-dimensional, inherently buttressed ice-sheet-shelf system to investigate the ice-sheet response to a simultaneous perturbation in snowfall and sub-ice-shelf melting. The perturbation is mass preserving, i.e., the amount of prescribed additional mass gain through snowfall and mass loss through basal melting, respectively, are equal (see Fig. 1). We find that under such forcing an ice sheet can undergo retreat while gaining mass, a behavior which is robust under variation of relevant parameters affecting the bed geometry and ice dynamics. To investigate

the influence of the magnitude and the pattern of sub-ice-shelf melting we carry out the parameter ensemble for a range of perturbation strengths and apply melting to different regions of the ice shelf (central vs. lateral regions) and for different degrees of distribution. The short-term and long-term ice-sheet response is analyzed with respect to changes in ice discharge, grounding-line position and ice-shelf backstress. Furthermore, we examine the relative importance of changes in ice-shelf length vs. changes in ice-shelf thickness on the response in ice-shelf backstress.

## 2   Methods

### 2.1   Model setup

We use the open-source Parallel Ice Sheet Model (PISM; Bueler and Brown, 2009; Winkelmann et al., 2011; The PISM authors, 2018), version stable07 (https://github.com/pism/pism/). The model applies a superposition of the shallow-ice approximation





(SIA; Morland, 1987) and the shallow-shelf approximation (SSA; Hutter, 1983) of the Stokes stress balance (Greve and Blatter, 2009). In particular, the SSA allows for stress transmission across the grounding line and thus accounts for the buttressing effect of laterally confined ice shelves on the upstream grounded regions (Gudmundsson et al., 2012; Fürst et al., 2016; Reese et al., 2018). The model applies a linear interpolation of the freely evolving grounding line and accordingly interpolated basal

friction, and uses one-sided differences in the driving stress close to the grounding line (Feldmann et al., 2014). Basically, the model is set up and spun up according to the MISMIP+ design (e.g., model parameters, ice rheology, basal friction law, three-dimensional model domain, channelized flow; see Asay-Davis et al., 2016, for details). In particular, this includes the assumption of isothermal ice (ice softness $A$ constant in space and time). Further, during ice-sheet spinup the rate of surface accumulation $a$ is constant in space and time and sub-ice-shelf melting is zero ($m = 0$). Values of the relevant parameters

varied throughout the study are given in Table 1.

The three-dimensional setup is designed to model a marine ice sheet, which drains through a bed trough, feeding a bay-shaped ice shelf which calves into the ocean. The idealized bed topography (Fig. 4b) is a superposition of two components and very similar to the one used in (Feldmann and Levermann, 2017): the bed component in x direction, $B_x(x) = -150\,\mathrm{m} - s \cdot 0.9 \cdot 10^{-3}|x|$, with bed-slope scaling factor $s$, is an inclined plane sloping down towards the ocean (Fig. 4a). In contrast to

Asay-Davis et al. (2016) the bed geometry prescribed here thus does not have an overdeepening (oceanward up-sloping bed section). The component in y direction, $B_y(y)$, is the one used in the MISMIP+ experiments which has a channel-shaped form (Asay-Davis et al., 2016, see their Eq. 3 and Fig. 1b). The superposition of both components yields a bed trough which is symmetric in both x and y directions (symmetry axes $x = 0$ and $y = 0$). While the main ice flow is in x direction (from the interior through the linearly sloping bed trough towards the ocean) there is also a flow component in y direction, i.e., from

the channel's lateral ridges down into the trough. Resulting convergent flow and associated horizontal shearing enable the emergence of ice-shelf buttressing, which leads to a grounding-line position further downstream than in the absence of an ice shelf. Ice is cutoff from the ice shelf and thus calved into the ocean beyond a fixed position (Fig. 4). Due to the symmetry of the setup, throughout our analysis we only consider the right-hand half of the domain, which has a length of $700\,\mathrm{km}$ (x direction) and a width of $80\,\mathrm{km}$ (y direction). The simulations are carried out using a horizontal resolution of $1\,\mathrm{km}$.

## 2.2   Experimental design

Using the model setup described above the simulations are initiated with a block of ice from which a marine ice-sheet-shelf system evolves. While the model spinup is closely along the lines of the MISMIP+ experiments (see Sect. 2.1), the design of the perturbation in this study is completely different. Starting from the steady-state ice sheet, mass-redistribution experiments are carried out applying a twofold perturbation by introducing 1) basal melting to the floating ice shelf and 2) additional snowfall

to the grounded ice sheet. This perturbation is designed such that the applied forcing is mass preserving, i.e., the rate of mass of additional precipitation, $\widetilde{a}$, is equal to the rate of ice mass molten at the underside of the ice shelf, $\widetilde{m}$, and thus $\widetilde{a} = \widetilde{m}$ (Fig. 1). It is referred to as "mass redistribution" experiment in the following. Sub-ice-shelf melting is distributed over an area that follows the grounding line if it migrates while the extent of that area (number of grid cells), $A_{\widetilde{m}}$, is constant in time. The additional snowfall is confined to a fixed area in the interior of the model domain of extent $A_{\widetilde{a}}$, ensuring that the entire ice





mass is added upstream of the grounding line considering the possibility of strong grounding-line retreat (Fig. 2a). An initial examination of the ice-sheet response to changes in $A_{\widetilde{a}}$ revealed only minor differences in the results and thus this parameter is not varied in the present study. In order to investigate the effects of the two different types of forcing separately, we also carry out perturbation experiments with only one of the two forcings in effect, termed as "melting only" ($\widetilde{a} = 0$) and "accumulation

only" ($\widetilde{m} = 0$).

The mass redistribution experiments are conducted for two different regions of applied sub-shelf melting, termed as "central melting" and "lateral melting". Central melting is confined to the center of the ice shelf, where the ice stream crosses the grounding line (Fig. 2b). Lateral melting is applied to the two lateral parts of the ice-shelf bay where the ice flows from the ridges into the ice shelf. In each of the two parts melting is distributed over an area $A_{\widetilde{m}}/2$ (Fig. 2c). Both types of perturbation

are symmetric to the centerline of the setup. The location of the melt area $A_{\widetilde{m}}$ is determined at each model time step and hence adapts to grounding-line movement. It includes the first floating grid cells directly downstream of the grounding line and extends a given number of grid cells into the ice shelf; in x-direction for "central melting" and in y-direction for "lateral melting", respectively. For "central melting", the perturbation area extends between $y = \pm 20$ km along the grounding line, making sure that no lateral parts of the grounding line are included. In the "lateral melting" experiment, the perturbation area

is determined from the extent of the confined ice shelf, i.e., the x-location of the grounding line at the center ($y = 0$ km) and at the margins ($y = \pm 40$ km) of the channel setup. The perturbation area extends 20 km along the lateral grounding line, with the center shifted slightly upstream to exclude melting near "fangs" – grounded features between 480 and 510 km (Asay-Davis et al., 2016). The extent, i.e., the width, of the perturbation area $A_{\widetilde{m}}$, is varied in different experiments to compare between very confined and more distributed melt patterns while keeping the total sub-shelf mass flux constant.

Further parameters varied include the bed slope $s$, the width of the bed channel $w_c$, the ice softness $A$ and the baseline accumulation rate $a$. Since a variation of these parameters alters the equilibrium ice-sheet geometry and the flow dynamics, separate ice-sheet spinups are carried out by varying one single parameter of $A$, $s$, $a$ and $w_c$ at a time. This way a set of nine equilibrium runs is created (see Table 1 for parameter sampling), providing the initial steady states for the respective perturbation experiments. Forcing this set of equilibria by basal melting in the two different ice-shelf parts (central and lateral

parts), using three different melt areas $A_{\widetilde{m}}$, creates an ensemble of $9 \cdot 2 \cdot 3 = 54$ simulations. This ensemble is run for different magnitudes of perturbation, ranging from $\widetilde{a} = \widetilde{m} = 0.5$ to 6 Gt/yr.

## 3   Results

Spinning up the model, a marine ice-sheet-shelf system evolves, with channelized ice-stream flow through the bed trough. The ice sheet feeds a bay-shaped ice shelf which provides backstress on the upstream grounded ice sheet and calves into the

ocean. Reaching equilibrium, the surface flux over the grounded ice sheet (mass influx through snowfall) is in balance with the grounding-line flux (mass discharge across the grounding line), yielding an ice sheet with constant volume and a stable grounding line position (Fig. 3). Perturbing this steady state by forcings of (1) sub-ice-shelf melting and (2) surface accumulation the ice-sheet-shelf system evolves towards a new equilibrium. To analyze the effects of these two forcings separately,



first we consider experiments which apply only one forcing (increased accumulation *or* sub-shelf melting) before analyzing the mass redistribution experiments (combining both forcings in a mass-preserving way). The results are discussed in the following three subsections addressing the case of central melting. The response to the lateral melt perturbation is qualitatively very similar (compare Figs. 3 and 4 to Figs. A5 and A6), but differs quantitatively from the central melting.

## 3.1 "Melting only" perturbation

Prescribing non-zero sub-shelf melting ($\widetilde{m} > 0$) while not altering the surface accumulation ($\widetilde{a} = 0$, first column in Fig. 3) leads to ice-shelf thinning and hence reduces the backstress of the ice shelf on the grounded ice sheet. In turn, this causes speed-up and thinning of the grounded portion of the ice sheet, inducing grounding-line retreat. On the long term, the thinning propagates several $100 \, \mathrm{km}$ upstream, reaching the ice divide (Fig. 4a). Approaching a new equilibrium, the perturbed ice sheet has retreated and thinned, both resulting in a loss in ice volume (Fig. 3c).

From a flux-point-of-view the new steady state is reached when the flux across the grounding line (ice discharge) has adjusted to the decrease in surface flux over the grounded ice sheet (due to grounding-line retreat) and both are in balance again. Immediately after the perturbation onset the grounding-line flux increases due to the acceleration of ice flow before it peaks and declines towards the grounded surface flux (Fig. 3b). The cumulative additional discharge during this period of flux imbalance determines the amount of the eventual ice-volume loss.

## 3.2 "Accumulation only" perturbation

Increasing the surface accumulation over the grounded portion of the ice sheet ($\widetilde{a} > 0$) while leaving sub-shelf melting at zero ($\widetilde{m} = 0$, second column in Fig. 3) leads, as expected, to a thickening and an advance of the ice sheet (Fig. 4b). On the short term, the prescribed step-shaped increase in grounded surface accumulation and a further sub-sequent small increase due to grounding-line advance is undercompensated by the grounding-line flux (Fig. 3b). On the long term, the grounding-line flux adjusts and increases gradually, eventually balancing the surface mass flux. Reaching a new equilibrium, the grounding line has advanced and the ice sheet gained volume (Fig. 3c). In contrast to the sub-shelf-melting perturbation, the accumulation perturbation has no direct effect on the ice-shelf backstress though there are indirect effects including 1) ice-shelf thickening (through the downstream propagation of ice into the ice shelf) and 2) shortening of the ice shelf (due to grounding-line advance).

## 3.3 "Mass redistribution" perturbation

The surface accumulation perturbation and the sub-shelf melting perturbation are combined in a mass-preserving way ($\widetilde{a} = \widetilde{m}$, as described in Sect. 2.2). The most frequent ice-sheet response within the ensemble is characterized by grounding-line retreat which is accompanied by an increase in ice volume (Fig. 3, third column), termed as retreat-volume-gain response in the following. Similar to the "accumulation only" perturbation here the step-shaped increase in surface accumulation results in a transient imbalance between the grounded surface flux and ice discharge across the grounding line during which the grounded ice sheet gains mass. At the same time, the applied melting beneath the ice shelf causes grounding-line retreat as seen in the



"melting only" experiment. Ice speed-up, associated increase in grounding-line flux and grounding-line retreat, all going along with the melt perturbation, can compensate the forced increase in surface accumulation on the long term, effecting a smaller volume increase compared to the "accumulation-only" experiment.' The thinning of the grounded ice sheet remains confined to an area of a few 10 km upstream of the grounding line (which is about 10 % of the ice-sheet length) and grounding-line

retreat is weaker than in the "melting only" experiments (Fig. 4). The contributions from the two stand-alone perturbations to changes in grounded volume and grounding-line position, respectively, sum up to yield the result of the combined forcing experiment (third row in Fig. 3).

The retreat-volume-gain response is found for both applied melting patterns, i.e., central and lateral melting (Fig. 5). For the case of central melting, this response most frequently occurs for a very small melting area ($A_{\widetilde{m}} = 200$ km$^2$, covering $\sim 10$ %

of the ice-shelf confinement area) and a medium to large perturbation magnitude ($\geq 3$ Gt/yr, indicated by ensemble mean). For a lower perturbation strength the grounding line undergoes a slight advance instead of retreat. The change in grounded ice mass increases sub-linearly with perturbation magnitude and is positive for all ensemble members. A comparison to the most distributed central melting pattern ($A_{\widetilde{m}} = 600$ km$^2$) reveals a shift of the advance/retreat threshold of the grounding line towards a larger perturbation magnitude of about 4 Gt/yr (Fig. 6). Considering lateral melting, the retreat-volume-gain

response applies to the vast majority of the analyzed perturbation magnitudes. An exception from this behavior occurs for the case of strong, distributed melting, where, instead of monotonous increase, the ice-volume gain peaks before declining towards zero as the grounding-line retreat increases superlinearly (compare Fig. 5 to Fig. 6). This is the turning point where backstress reduction starts to become as important in controlling grounding-line position and flux as snowfall increases are. Note that due to the comparatively low advection of ice from the lateral ridges into the ice shelf, a lateral melt perturbation of sufficient

strength can melt away the ice shelf locally and the corresponding simulations are discarded from the analysis due to lacking comparability.

### 3.4 Short-term response to central vs. lateral melt

To investigate the short-term dynamic response of the ice sheet to changes in basal melting in two very different ice-shelf regions (central vs. lateral parts, Fig. 2), the change in ice discharge integrated along the grounding line is evaluated throughout

the ensemble for the first few ten model years. To ensure comparability, melting is applied over an equal area of $A_{\widetilde{m}} = 400$ km$^2$ for the two melt patterns.

The quasi-instantaneous response magnitudes (after ten years) are very similar between the two applied melt patterns (Fig. 7a). Considering the mean values for each melt magnitude, there is a slightly stronger increase in grounding-line flux in the case of central melting. However, given the relatively large spread of results coming out of our individual simulations

this difference is not significant. After about 50 model years the results from the two sets of experiments have become clearly distinguishable. Now the response behavior has flipped, i.e., lateral melting induces a significantly stronger flux response than central melting (Fig. 7b). In other words, within a few ten years the grounded ice loss becomes much more sensitive to melting in the lateral parts than to melting in the central part of the ice shelf.





This transition can be explained by our finding that basal melting in the lateral ice-shelf regions has a larger effect on local ice-shelf thinning since the ice supply from the lateral ridges is comparatively low. Consequently, continuous lateral melting causes a strong backstress reduction, resulting in a substantial grounding-line flux increase. In contrast, for the central melt case much of the original ice-shelf thickness in the melting area is maintained also for the highest considered melt magnitudes due to stronger advection of ice through the bed channel into the central grounding-line region. This way the deviation in response between the two perturbation patterns increases with time and melt magnitude, although both patterns apply the same rate of mass loss distributed over an equal area of the ice shelf (Fig. 7).

High cumulative lateral melting (due to strong, long and/or confined melt) favors local ice-shelf melt-away and in case of occurence the corresponding simulations are excluded from the analysis (see also end of Section 3.3). This lack of data points makes an investigation of the flux response later than 50 yr suffer from analyzability, especially for large melt rates. The choice of a medium-sized melt area of $A_{\widetilde{m}} = 400\ \mathrm{km}^2$ taken here reduces local melt rates compared to the most confined case ($A_{\widetilde{m}} = 200\ \mathrm{km}^2$), allowing the above comparison between central and lateral melt within the first 50 model years also for high perturbation strenghts. At the same time the two regions of applied melt are still well distinguishable as there is only little overlap between the them (compare Figs. 2b and c).

## 3.5   Quantifying backstress sensitivity to changes in ice-shelf geometry

It has been shown in idealized experiments that the degree of ice-shelf backstress strongly depends on the geometry of the ice shelf and its confinement, i.e., the ice-shelf thickness as well as the width and the length of the ice-shelf confinement (Dupont and Alley, 2006; Goldberg et al., 2009; Gudmundsson et al., 2012). The perturbations applied throughout our experiments modify these ice-shelf characteristics, therefore affecting the ice-shelf backstress. Since the ice-shelf width is strongly confined by the side walls of the prescribed bed trough in our simulations, changes in the width of the ice shelf in the course of the perturbation are negligible, irrespective of the perturbation magnitude. Consequently, here we assume that the changes in ice-shelf length and thickness are the dominant drivers of changes in the backstress.

In the following we carry out a quantification of the backstress sensitivity to ice-shelf length and thickness changes which holds for the particular characteristics of the channel-type ice-sheet-shelf system considered in this study. To provide a set of different equilibrium ice-shelf backstress configurations (determined by the equilibrium ice-shelf geometry) we exploit the entire ensemble of simulations (including variations of bed slope, channel width, ice softness, baseline accumulation rate, melt-perturbation pattern and area, see Table 1), analyzing the individual responses to each of the three forcing types ("melt only", "accumulation only", "mass redistribution").

In order to quantify the contributions of changes in ice-shelf length, $\Delta L$, and ice-shelf thickness, $\Delta H$, respectively, to changes in ice-shelf backstress, $\Delta\Theta$, in our simulations, we assume a relation of the following form:

$$\Delta\Theta = A_{\Delta L}\Delta L + B_{\Delta H}\Delta H \quad \text{with} \quad A, B > 0. \tag{1}$$

In other words, we linearize the dependency of $\Delta\Theta$ on $\Delta L$ and $\Delta H$ for small perturbations. This linear approach is justified by the model results to a good approximation as exemplarily shown in Figs. A1 and A2. The three variables $\Delta\Theta$, $\Delta L$ and





$\Delta H$ are scalars diagnosed from the simulated steady states with $\Delta$ referring to the long-term change (new equilibrium after perturbation) normalized to the initial equilibrium value, making Eq. (1) dimensionless (see Appendix A1 for details). The weights $A_{\Delta L}$ and $B_{\Delta H}$ quantify the sensitivity of ice-shelf backstress to changes in ice-shelf length and ice-shelf thickness, respectively. To be able to obtain the values of $A_{\Delta L}$ and $B_{\Delta H}$ for each parameter combination and melt pattern considered in

this study (see Sec. 2.2) we carry out the three different forcing types ("accumulation only", "melt only", "mass redistribution") for the entire ensemble. The method to infer these numbers from our simulations is detailed in Appendix A1. A qualitative discussion of the physical relations between changes in ice-shelf length/thickness and ice-shelf backstress for each of the three different forcing types is given in Appendix A2.

    The results from our quantification reveal that the ice-shelf backstress is generally more sensitive to changes in ice-shelf

thickness than in ice-shelf length. This outcome is independent of the prescribed parameter combination and the location and spatial extent of the perturbation ($A_{\Delta L} < B_{\Delta H}$ for the entire ensemble of all simulations carried out in this study, Fig. 8). The ratios range from $A_{\Delta L}/B_{\Delta H} = 0.3$ to $0.97$, while the vast majority of the conducted ensemble of simulations lies between $A_{\Delta L}/B_{\Delta H} = 1/4$ and $2/3$ (except for three out of the 54 ensemble members, grey-shaded area in Fig. 8). This means that ice-shelf thickness changes generally are about 1.5 to 4 times as effective as changes in ice-shelf length in terms of backstress

modification. In other words, a 10 % increase/decrease of ice-shelf thickness would require a 15 to 40 % increase/decrease in ice-shelf length in order to yield the same amount of backstress change. The experiments with prescribed lateral ice-shelf melt all yield ratios of $A_{\Delta L}/B_{\Delta H} < 0.5$ (dashed line in Fig. 8). The central melt perturbation yields relatively low values of $A_{\Delta L}$ and $B_{\Delta H}$ (data points concentrated in the lower-left corner of Fig. 8) while the lateral melt perturbation results in much higher values, which reveals that the backstress changes generally are more sensitive to lateral than to central melting,

respectively. This fits to our results from above which also revealed a higher sensitivity to lateral melt in terms of the long-term and short-term ice-sheet response, i.e., changes in grounding-line position, ice volume and ice discharge, which all are induced by backstress changes. Furthermore, more distributed lateral melt leads to slightly higher $A_{\Delta L}/B_{\Delta H}$ ratios compared to confined melting, which is vice-versa for the case of central melt (colors in Fig. 8).

## 4   Discussion and conclusions

Carrying out idealized simulations of an inherently buttressed, marine ice-sheet-shelf system we show that applying a mass-preserving forcing can lead to a retreating but mass-gaining ice sheet. Mass preservation of the forcing is realized by applying melting to the underside of the floating ice shelf while adding an equal amount of mass as snowfall on top of the interior part of the grounded ice sheet (Figs. 1 and 2). We investigate the ice-sheet response for an ensemble of simulations spanned by varied model parameters as well as different patterns and magnitudes of sub-ice-shelf melting.

### 4.1   Long-term response

The grounding-line retreat in the retreat-mass-gain response is induced by a reduction in ice-shelf backstress, which is driven by the applied sub-shelf melting. The associated ice speed-up upstream of the grounding line causes a significant increase in



ice discharge across the grounding line (Fig. 3b). However, these changes cannot compensate for the additional mass influx due to the applied increase in surface accumulation, leading to a net mass gain of the grounded ice sheet (Fig. 3c). Vice versa, the surface perturbation cannot make up for the grounding-line retreat, induced by the sub-shelf melting perturbation, resulting in a response of simultaneous grounding-line retreat and ice-sheet volume gain (Fig. 4). We would like to note that in the model

the calculation of the driving stress (surface slope) close to the grounding line separates between grounded and floating boxes: instead of a centered difference scheme across the grounding line, two one-sided differences upstream and downstream of the grounding line are used to compute the driving stress in the last grounded and first floating cell, respectively (Feldmann et al., 2014). Consequently, a possible direct effect of ice-shelf thinning close to the grounding line on the upstream driving stress of grounded ice as mentioned in Rignot (2002) and Winkelmann et al. (2012) can be excluded in our simulations. Thus, changes

in grounded ice velocities, ice thickness and grounding-line position result solely from changes in ice-shelf backstress.

The retreat-mass-gain response occurs for the majority of the conducted ensemble simulations, i.e., for different channel widths, bed slopes, baseline surface accumulation rates and ice softness values (Table 1). The ensemble is carried out for two different regions of applied sub-ice-shelf melting (central and lateral ice-shelf areas) as well as for different extent of the areas over which the mass loss is spread (Fig. 2). Generally, grounding-line retreat is favored by strong melting and a high

concentration of melting close to the grounding line (Figs. 5 and 6). The case of more distributed melting allows to analyze larger lateral melt magnitudes, yielding the strongest grounding-line retreat magnitudes throughout the ensemble, accompanied by a significant reduction in ice-sheet volume gain. This reduction presumably occurs due to the substantial loss in grounded area. In fact, it is a general outcome of the ensemble simulations that for a given perturbation magnitude, volume gain is smaller for stronger grounding-line retreat. Also, comparing confined to distributed melting in our simulations, i.e., subtracting

the confined from the distributed results, reveals positive shifts in both grounding-line position and ice-volume for a given melt magnitude (compare Fig. 5 to Fig. 6). This is in agreement with results from idealized flowline simulations of a buttressed ice-sheet-shelf system by Gagliardini et al. (2010). They find grounding-line advance accompanied by volume gain when reducing the concentration of sub-ice-shelf melting to the grounding line, while leaving constant the total amount of melted ice. Exceptions from the retreat-mass-gain response are found for some cases of weak to medium central melting, in particular

if it is more distributed (slight grounding-line advance instead of retreat) and for the strongest magnitude of distributed lateral melt (decrease in grounded ice volume instead of increase for some ensemble members).

Stand-alone perturbations using only one of the two forcing components lead to more ordinary situations, i.e., grounding-line retreat going along with mass loss (only sub-shelf-melting perturbation applied, first column in Fig. 3) or grounding-line advance accompanied by mass gain (only surface accumulation perturbation applied, second column in Fig. 3), respectively.

A comparison to the mass redistribution case reveals that the simulated long-term changes in grounding-line position and grounded ice volume of the melting only and accumulation only cases superpose to a good approximation.

## 4.2 Short-term response

The short-term response of the ice sheet to central versus lateral melting depends on the duration of the applied melting. While the quasi-instantaneous change in grounding-line flux is slightly (but not significantly) larger for the case of central melting





(response after 10 years given in Fig. 7a), we find that for continued melting the total flux response becomes significantly more sensitive to lateral melt (response after 40 years given in Fig. 7b). We attribute this to the higher thinning rates in the case of lateral melting, which is applied over a region of relatively stagnant ice flow. Reese et al. (2018) find highest, instantaneous responses in grounded ice flux for thinning downstream of Antarctic ice streams (comparable to the central melt region here),

which is in line with the response to central melting being slightly higher after a few model years. They argue that the response is an increasing function of thinning, consistent with the signal from lateral melting to increase over the one from central melting within a few decades. Our results hence underline the important role of ice dynamics in the regions adjacent to the grounding line (grounded and floating regimes) interacting with enhanced sub-shelf melting to regulate grounded mass loss.

### 4.3   Quantifictaion of backstress sensitivity

Analyzing the influence of ice-shelf changes on the backstress of the ice shelf exerted on the upstream ice sheet, we find that changes in ice-shelf thickness are more relevant than changes in ice-shelf length which holds for the entire ensemble of simulations (Fig. 8). For the vast majority of the ensemble, ice-shelf thickness changes have about 1.5 to 4 times more effect on the backstress than ice-shelf shortening has. We also infer that backstress changes are generally larger for lateral melting compared to central melting, which is consistent with a higher lateral-melt sensitivity in terms of ice loss and grounding-

line retreat discussed above. These results are limited to the (geometric) characteristics of the channel-type ice-sheet-shelf systems considered in this study. Different initial ice-shelf geometries and thus different initial ice backstress conditions are accounted for by the conducted ensemble of simulations, in particular through the modification of the prescribed bed-channel geometry and ice properties. Our approach to quantify backstress sensitivity assumes that changes in backstress depend on both thickness change and length change (Dupont and Alley, 2006; Goldberg et al., 2009) in a linear way and neglects other

possible parameters. Furthermore, changes in ice-shelf length or thickness are claimed to depend linearly on the perturbation magnitude (Appendix A1). Consequently, our assumptions mean a strong simplification to the complex problem of ice-shelf backstress. At the same time, the applied linearization is supported by the data obtained from our simulations (Figs. A1 to A4). The results coming out of this first-order approach are robust and reveal a consistent picture when comparing the relevance of changes in ice-shelf thickness versus length.

### 4.4   Assumptions and idealizations in model setup and experiments

There are several simplifications in the design of the model setup and the experiments (shallow stress balance, isothermal ice, idealized bed topography and perturbation, fixed calving front), thus reducing complexity of modeled ice flow. At the same time our approach allows for an analysis focused on the essential effects of perturbations to the ice surface and the ice-shelf bottom (and their interaction) on the ice dynamics, isolated from unwanted (secondary) effects that would result from a more

complex model realization, while our simulations still incorporate the relevant physics of ice flow.

The synthetic bed topography and the idealized forcing used here aim at a conceptual understanding of the ice-sheet response to the applied perturbation in contrast to the attempt of investigating a real-world system that would include a much wider range of physical effects. For instance, the smooth bed geometry used here does not account for bumps usually found in observations



of the sub-glacial topography and which would interfere with grounding-line dynamics (e.g., Alley et al., 2007; Favier et al., 2012). Also, the monotonously, landward upsloping shape of the bed prescribed here does not allow for an investigation of a Marine Ice Sheet Instability (Weertman, 1974; Schoof, 2007) which possibly plays a major role in the current evolution of the West Antarctic Ice Sheet (e.g., Joughin et al., 2014; Favier et al., 2014; Feldmann and Levermann, 2015; Seroussi et al., 2017).

Furthermore, the distribution of the surface mass balance and the sub-shelf melting in space and their evolution in time would be much more complex in a real-world system (e.g., Dutrieux et al., 2013; Frieler et al., 2015a) in contrast to the spatially very confined, step-like perturbations to the very simple equilibrium fields applied here. However, the approach taken here enables a simple and easily accessible implementation of a mass-preserving forcing and allows for an analysis of the first-order effects on ice discharge, grounding-line migration and ice-shelf backstress.

Comparing the magnitude of the surface accumulation perturbation in our simulations with estimates from paleo simulations and model projections for the Antarctic Ice Sheet (snowfall increase by $4-6\,\%\,\mathrm{K}^{-1}$, see Frieler et al., 2015a) yields that our experiments with lower to medium additional accumulation (about $7\,\%$ increase in total accumulation per $\mathrm{Gt}$ of perturbation) are in the range of possible Antarctic surface mass balance increase during the next century (Stocker et al., 2013). In the simplest case we can assume that (1) sub-ice-shelf melt rates are approximately linearly correlated to ocean temperatures,

increasing by $10\,\mathrm{m\,yr}^{-1}$ for each Kelvin, as estimated by Rignot (2002), and (2) ocean temperatures increase by about $0.1$ to $0.3\,\mathrm{K}$ per decade (supported by evidence in Schmidtko et al., 2014). Extrapolating this trend into the near future yields a possible increase of meltrates of several $10\,\mathrm{m\,yr}^{-1}$ within this century, which is consistent with the meltrate perturbations applied here (ranging from the orders of $\sim 1\,\mathrm{m\,yr}^{-1}$ to $\sim 10\,\mathrm{m\,yr}^{-1}$)'.

### 4.5   Relation to real world

Though the significance of our results is of course limited due to the idealized character of the conducted experiments, eventually we would like to relate them to real-world systems such as the outlets of the Antarctic Ice Sheet. We conclude that if a basal melt/accumulation pattern of increased snowfall and sub-ice-shelf melting, respectively, similar to the one used here, would apply under future warming, our findings suggest a tendency of ice-sheet outlets to retreat while becoming steeper. This reasoning is limited to glaciers with landward upsloping bed, as examined in our experiments, not accounting for the role of a

possible Marine Ice Sheet Instability. Whether such an ice sheet would indeed gain mass at the same time (as generally found in the present study) would depend on several factors, including the distribution and the magnitude of the snowfall increase (in relation to the basal-melt magnitude), which is not investigated in detail here. Furthermore, our results highlight the relevance of Antarctic ice-shelf thinning in regions of high lateral shearing (shear margins) where the ice shelf is fed from grounded regimes of relatively stagnant ice flow. Such ice-shelf regimes might be susceptible to substantial backstress reduction in case

of continuous local access of warm water.



**Table 1.** Parameters and their values varied throughout the experiments. Values applied in the default simulation are highlighted in bold.

| Parameter | Value | Unit | Physical meaning |
|---|---|---|---|
| $a$ | $\{0.15, \mathbf{0.3}, 0.45\}$ | $\mathrm{m\,yr^{-1}}$ | Baseline surface accumulation rate |
| $\widetilde{a}$ | $\{0, ..., \mathbf{6}\}$ | $\mathrm{Gt\,yr^{-1}}$ | Magnitude of accumulation perturbation |
| $A$ | $\{4 \cdot 10^{-25}, \mathbf{8 \cdot 10^{-25}}, 1.6 \cdot 10^{-24}\}$ | $\mathrm{Pa^{-3}\,s^{-1}}$ | Ice softness (Glen's flow law coefficient) |
| $A_{\widetilde{m}}$ | $\{\mathbf{200}, 400, 600\}$ | $\mathrm{km^2}$ | Area of sub-shelf melt perturbation (see Fig. 2) |
| $\widetilde{m}$ | $\{0, ..., \mathbf{6}\}$ | $\mathrm{Gt\,yr^{-1}}$ | Magnitude sub-shelf melt perturbation |
| $s$ | $\{\mathbf{1.0}, 1.1, 1.2\}$ | | Bed-slope scaling factor for bed component in x direction, $B_x(x)$ |
| $w_c$ | $\{\mathbf{24}, 28, 32\}$ | km | Half-width of bed trough (entering Eq. 4 of Asay-Davis et al., 2016) |

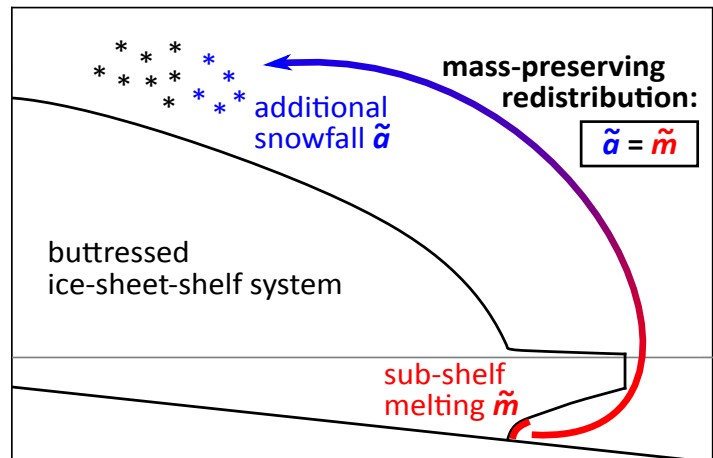

**Figure 1.** Schematic showing the two kinds of perturbations applied to the ice-sheet-shelf system in this study. The mass-redistribution experiments are designed in a mass preserving way, i.e., the rate of basal ice-shelf melting (in terms of mass loss per unit time) is equal to the rate of additional snowfall put on top of the ice-sheet interior.





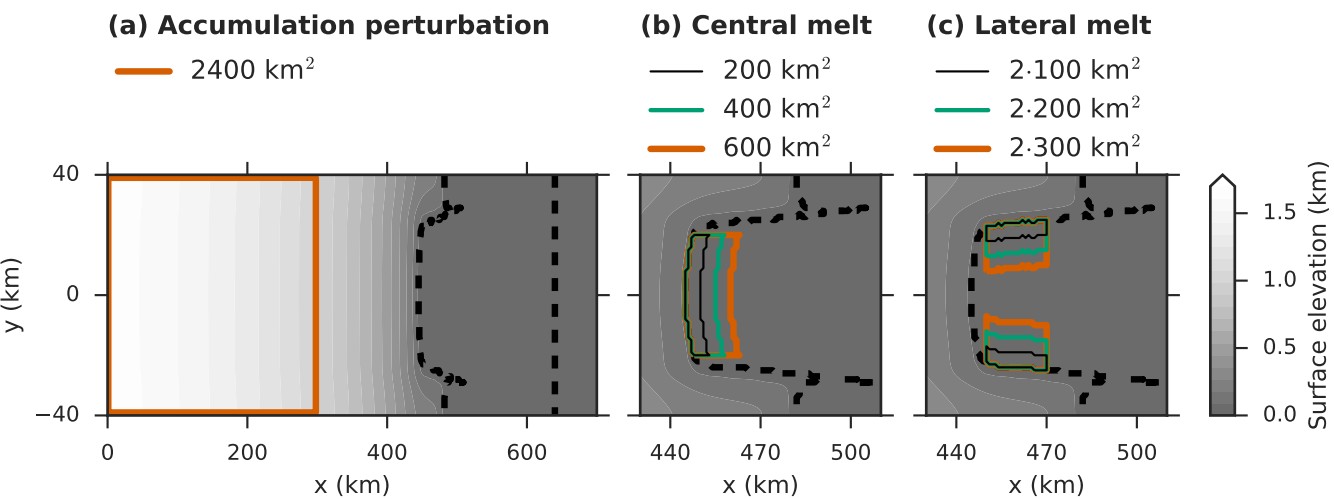

**Figure 2.** Regions and spatial extent of the perturbations applied throughout the different experiments (colored contours with ice-sheet surface elevation in the background). **(a)** Snowfall perturbation in the ice-sheet interior. Sub-ice-shelf melting is prescribed in **(b)** the ice-shelf center and **(c)** laterally (zoom into grounding-line region). Dashed contours show positions of steady-state grounding line before perturbation and the fixed calving front, respectively.





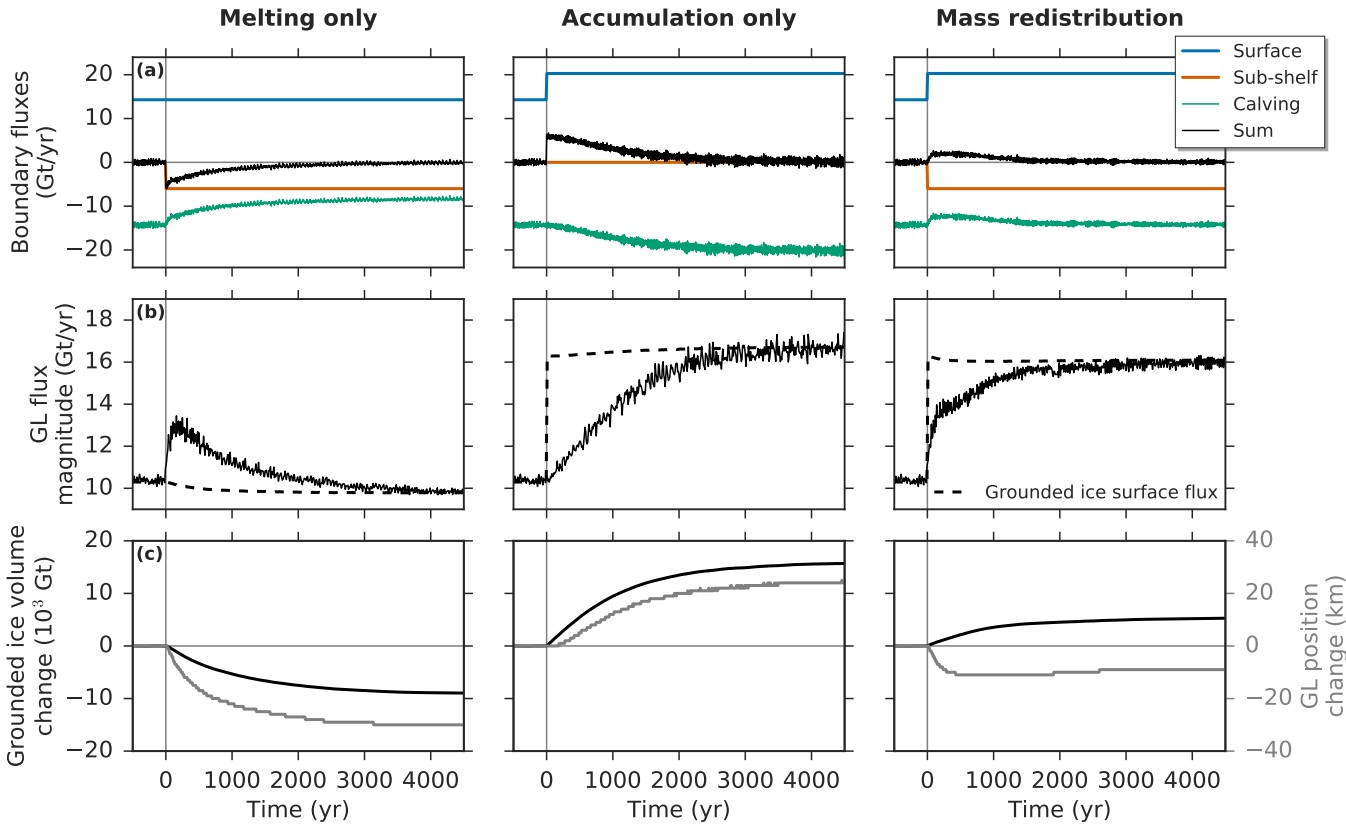

**Figure 3.** Timeseries for the three different types of forcing (columns) for the central melt experiment using the default set of model parameters (see Table 1). **(a)** Boundary fluxes of the ice-sheet-shelf system, including fluxes integrated over the ice surface (mass gain through accumulation), the ice shelf-bottom (mass loss through melting), the calving flux, as well as the sum of the three fluxes. **(b)** Flux across the grounding line (straight) and surface mass flux integrated over the grounded part of the ice sheet. **(c)** Grounded ice volume (black) and change in grounding-line position (grey, diagnosed at centerline of the domain). In all panels, the time axis has its zero point at the onset of the perturbation (vertical line) and also covers the last 500 yr of model spinup (equilibrium run).





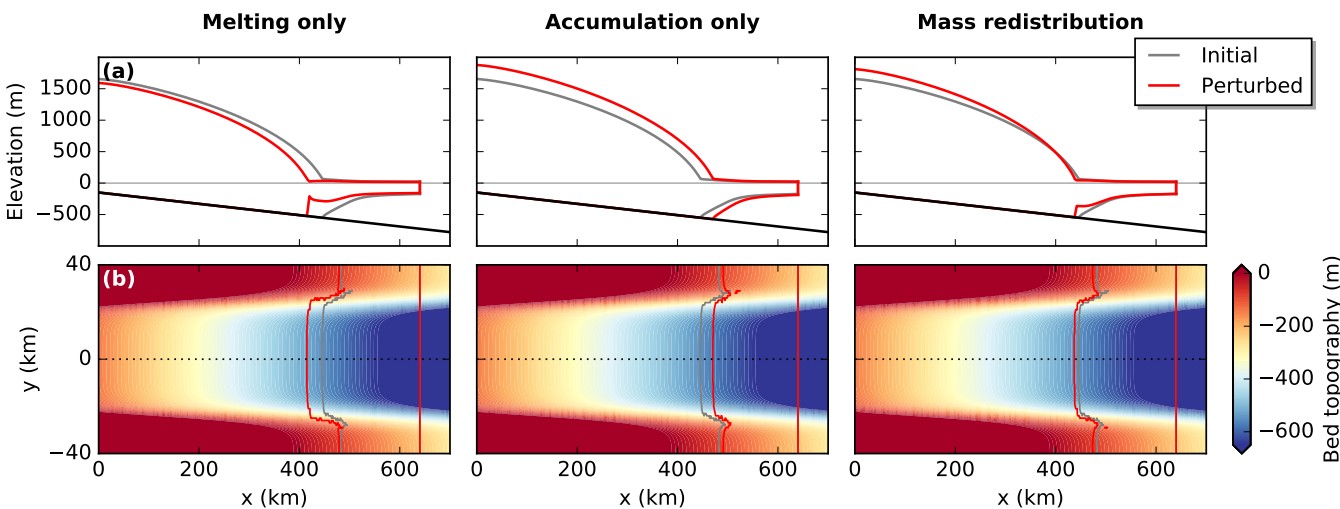

**Figure 4. (a)** Ice-sheet profiles along the centerline of the domain ($y = 0$) for the three different forcing experiments (columns) using the default set of model parameters (see Table 1). Initial equilibrium before perturbation (grey) and final steady state after perturbation (red). **(b)** Positions of grounding line and calving front for the entire model domain overlain on bed geometry. Dotted line marks the centerline of the setup.



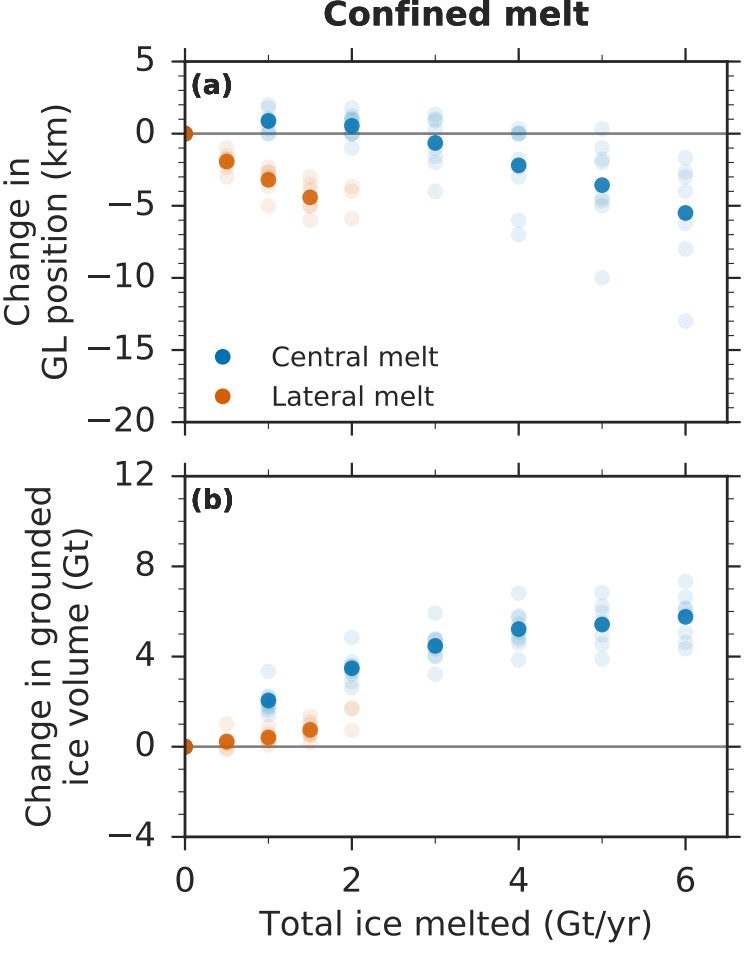

**Figure 5.** Long-term changes in **(a)** grounding-line position and **(b)** grounded ice volume for different magnitudes of confined sub-shelf melting, i.e., melting applied to a small area of $A_{\tilde{m}} = 200$ km$^2$ adjacent to the grounding line. Colors indicate location of the applied melt perturbation, i.e. central melt (blue) and lateral melt (orange) with areas indicated by black contours in Fig. 2b and c. Individual ensemble members are represented by transparent colors, average values for each perturbation magnitude and location are opaque.





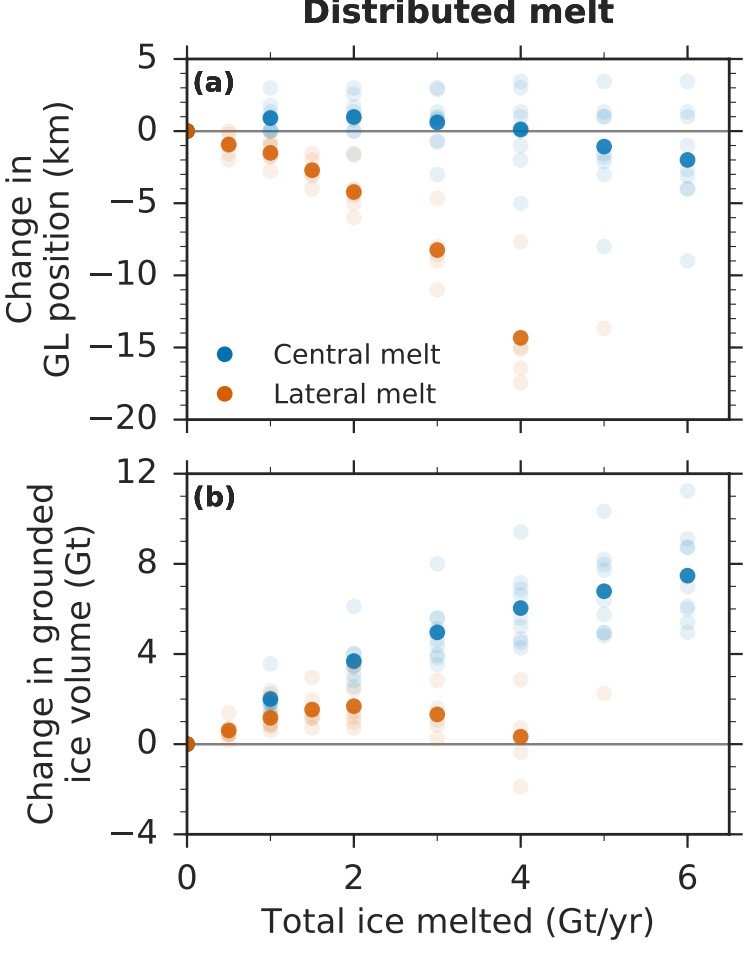

**Figure 6.** Long-term changes in **(a)** grounding-line position and **(b)** grounded ice volume for different magnitudes of distributed sub-shelf melting, i.e., melting spread over an area of $A_{\widetilde{m}} = 600 \text{ km}^2$ adjacent to the grounding line with areas indicated by orange contours in Fig. 2b and c. Colors same as in Fig. 5.





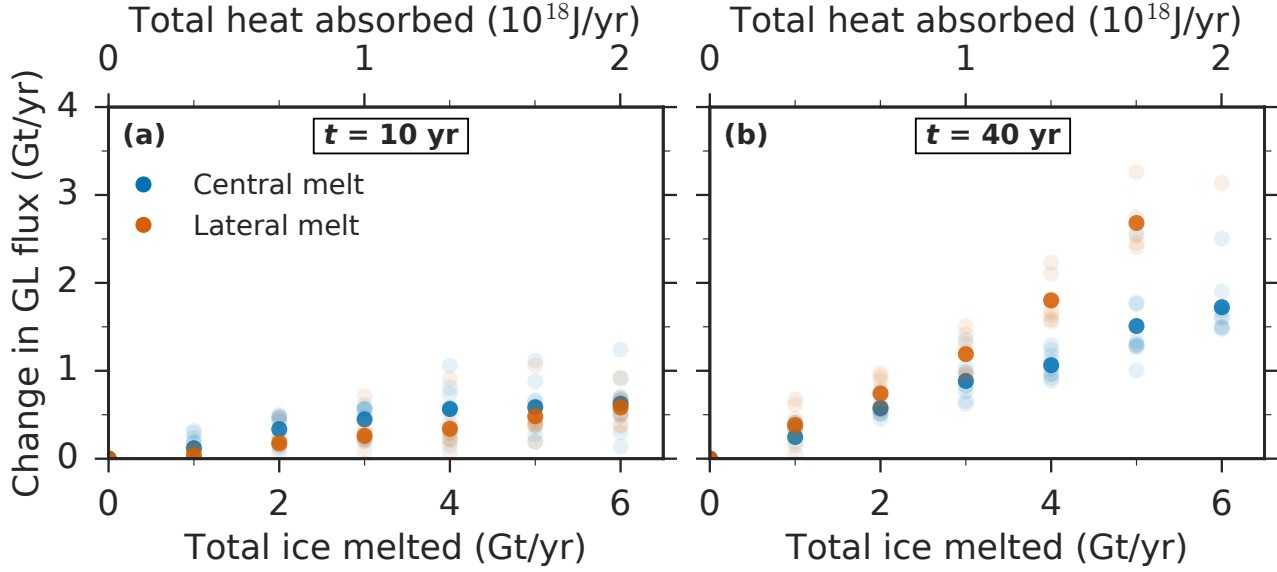

**Figure 7.** Increase in ice discharge across the grounding line after **(a)** 10 yr and **(b)** 40 yr of perturbation of central and lateral melt (blue and orange circles, respectively). Colors and markers same as in Figs. 5 and 6. Upper x-axis displays the amount of heat required to melt the amount of ice corresponding to the perturbation magnitude (lower x-axis).





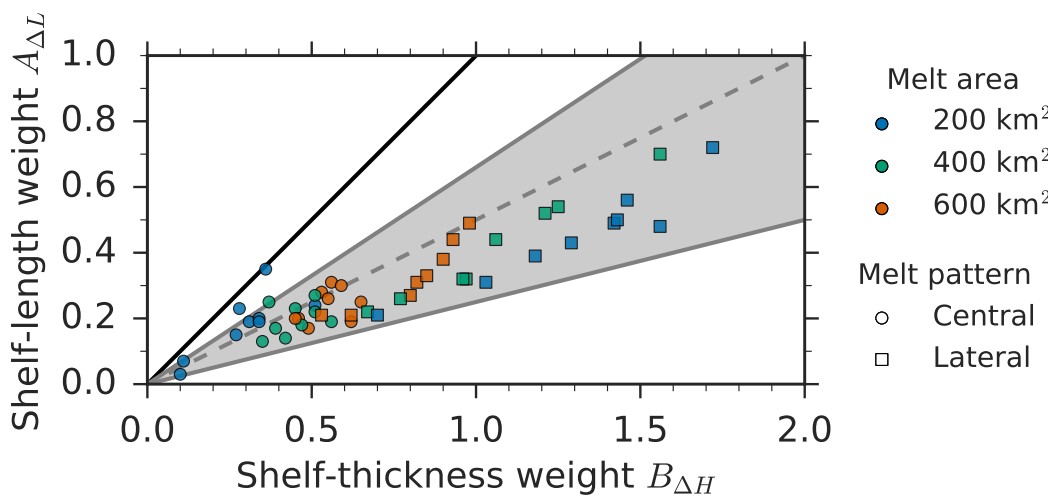

**Figure 8.** Values of ice-shelf-length weight $A_{\Delta L}$ versus ice-shelf-thickness weight $B_{\Delta H}$ (see Eq. 1) for the entire ensemble of simulations. Colors indicate extent of melt-perturbation area (increasing from blue to green to orange). Circles/squares denote experiments with applied central/lateral ice-shelf melting. Black line shows ratio $A_{\Delta L}/B_{\Delta H} = 1$, grey shaded area covering ratios between $A_{\Delta L}/B_{\Delta H} = 1/4$ to $2/3$ and thus edging the vast majority of the ensemble. Dashed grey line represents a ratio of $A_{\Delta L}/B_{\Delta H} = 0.5$, i.e., the upper bound for all lateral-melt experiments.





## Appendix A

### A1 Method to quantify backstress sensitivity to changes in ice-shelf geometry

We linearize the dependency of changes in the ice-shelf backstress, $\Delta\Theta$, on changes in ice-shelf length, $\Delta L$, and thickness, $\Delta H$, respectively, recalling Eq. (1) from Sec. 3.5:

$$\Delta\Theta = A_{\Delta L}\Delta L + B_{\Delta H}\Delta H \quad \text{with} \quad A_{\Delta L}, B_{\Delta H} > 0. \tag{A1}$$

Here $\Delta$ refers to the change normalized to the equilibrium value, making Eq. (1) dimensionless. Motivated by the fact that increasing ice-shelf thickness and/or ice-shelf length enhances ice-shelf backstress we expect the weights $A_{\Delta L}$ and $B_{\Delta H}$ both to be positive. We define the backstress-relevant ice-shelf length $L$ (confinement length) as the difference in grounding-line position at the side margin of the computational domain (representing the oceanward end of the confinement) and at the centerline (landward end of confinement, Fig. 2). The ice-shelf thickness $H$ is taken as the average over the area of this confinement. The backstress $\Theta$ is diagnosed at the central grounding line inside the bed trough, taken as the average of a $10\,\mathrm{km}$ long section which is symmetric around the centerline ($y = 0$).

The weights $A_{\Delta L}$ and $B_{\Delta H}$ are inferred for each member of the ensemble of simulations with its individual backstress configuration (determined by bed slope, channel width, ice softness, baseline accumulation rate, as well as perturbation area and pattern). For this purpose, we assume that length and thickness changes of the ice shelf each are linearly related to the perturbation magnitude $p$, thus

$$\Delta L = \alpha p \tag{A2}$$
$$\Delta H = \beta p. \tag{A3}$$

Insertion into Eq. (1) yields

$$\Delta\Theta = \gamma p, \tag{A4}$$

where $\gamma = \alpha A_{\Delta L} + \beta B_{\Delta H}$. To obtain $A_{\Delta L}$ and $B_{\Delta H}$ we apply the following procedure: First the coefficients $\alpha, \beta$ and $\gamma$ are inferred by applying linear fits according to Eqs. (A2-A4), see Figs. A1 and A2. This is done for each of the three different types of qualitatively very different forcings ("melt only", "accumulation only", "mass redistribution"), yielding three sets of coefficients, i.e., $(\alpha_i, \beta_i, \gamma_i)$ with $i = \{1, 2, 3\}$. These are used to solve the linear system

$$\gamma_i = A_{\Delta L}\alpha_i + B_{\Delta H}\beta_i, \quad i = \{1, 2, 3\}, \tag{A5}$$

for $A_{\Delta L}$ and $B_{\Delta L}$, respectively. Since the system of linear equations (Eqs. A5) is overdetermined and has no exact solution we compute the least-squares solution of the problem. We find the residuals to be negligible (at least one order of magnitude smaller than the solution). The resulting value pairs for $A_{\Delta L}$ and $B_{\Delta H}$ are visualized in a scatter plot to analyze the relative influence of changes in ice-shelf length and thickness, respectively, on ice-shelf backstress (Fig. 8).



## A2 Qualitative responses in ice-shelf geometry and backstress to three individual forcing types

In the following we analyze changes in ice-shelf length, thickness and backstress and their interrelation for each of the three applied forcing types based on the default set of model parameters. The responses of ice-shelf length $\Delta L$, thickness $\Delta H$ and backstress $\Delta \theta$ are to a good approximation linear in the perturbation magnitude $p$, with examples given in Figs. A1 to A4.

In the "accumulation only" experiments, the ice-shelf geometry is affected indirectly: The mass gains of the ice sheet lead to grounding-line advance and ice-shelf thickening ($\Delta H > 0$, blue data points in Figs. A1b to A4b), increasing the ice-shelf backstress. At the same time, the ice-shelf length decreases ($\Delta L < 0$, Figs. A1a to A4a), with a reducing effect on ice-shelf backstress. In the combination, the decrease in ice-shelf length has a larger imprint on backstress which linearly decreases with perturbation magnitude ($\Delta \theta < 0$, Figs. A1c to A4c). This effect is reversed in the "melting only" experiments. Here, grounding-

line retreat increases ice-shelf length ($\Delta L > 0$) and reduces thickness ($\Delta H < 0$) with an overall decrease in ice-shelf backstress ($\Delta \theta < 0$, orange data points).

In the "mass redistribution" experiments these effects act together. In most of the cases, the grounding line retreats and the ice-shelf thickness decreases ($\Delta H < 0$, black data points), similar to the melting only case. However, the response in the ice-shelf length is qualitatively different due to the additional snowfall which increases the ice flux mostly in the central bed trough

(ice stream), hampering grounding-line retreat in the center. Furthermore, the response depends on the spatial distribution of melting: For confined central melting, grounding-line retreat in the center and at the margins balance each other, as also the retreat in the stagnant lateral parts of the grounded ice sheet fringing the ice shelf is less pronounced, causing the ice-shelf length to remain approximately constant ($\Delta L \approx 0$, Fig. A1a). In contrast, for confined lateral melting, grounding-line retreat in the center is larger than at the margins causing the ice-shelf length to increase with perturbation magnitude ($\Delta L > 0$, Fig. A3a).

For distributed (central or lateral) melting the grounding-line retreat is stronger at the sides than in the center and the ice-shelf length overall decreases ($\Delta L < 0$, Figs. A2a and A4a).

*Competing interests.* The authors declare no conflict of interest.

*Acknowledgements.*





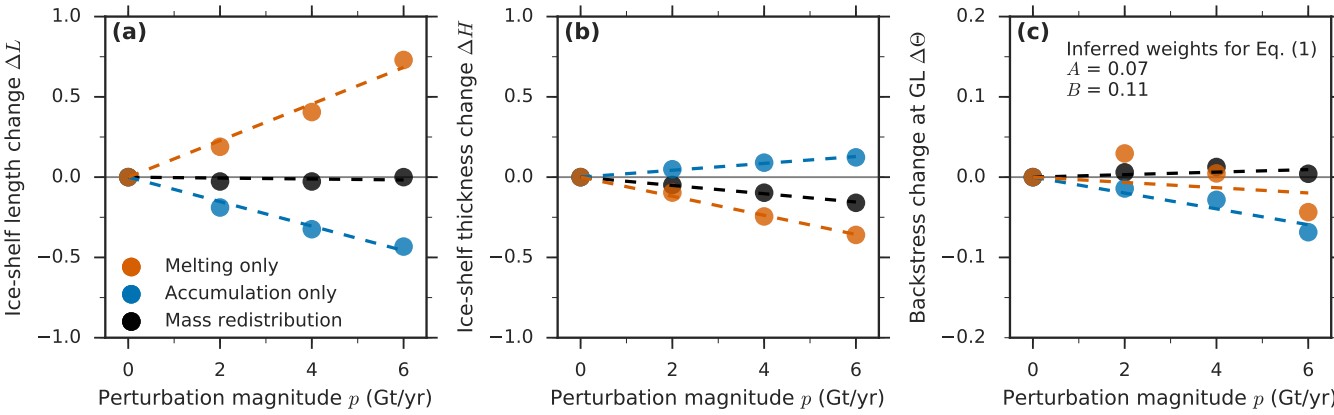

**Figure A1.** Linear regression of diagnosed normalized long-term changes in **(a)** length, **(b)** thickness and **(c)** backstress of the ice shelf in response to the three qualitatively different forcing types ("melting only" in orange, "accumulation only" in blue, "mass redistribution" in black), here shown exemplarily for the case of central melting and the default parameter set (very confined melting, i.e., $A_{\tilde{m}} = 200 \text{ km}^2$). Based on the inferred slopes (dashed lines) we are able to infer the backstress sensitivities $A$ and $B$ according to Eqs. (A1) - (A5).

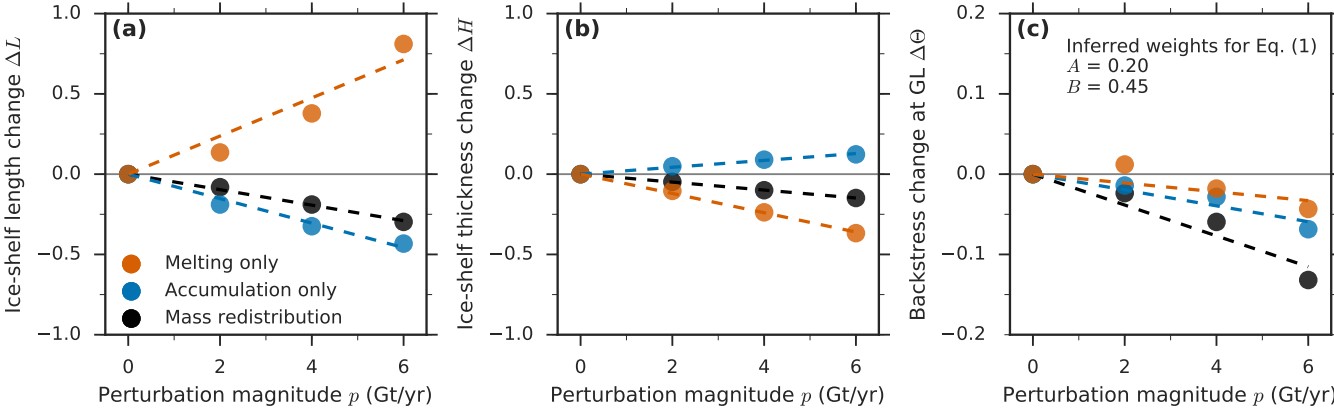

**Figure A2.** Linear regression as shown in Fig. A1 but here for the most distributed central melt pattern ($A_{\tilde{m}} = 600 \text{ km}^2$).



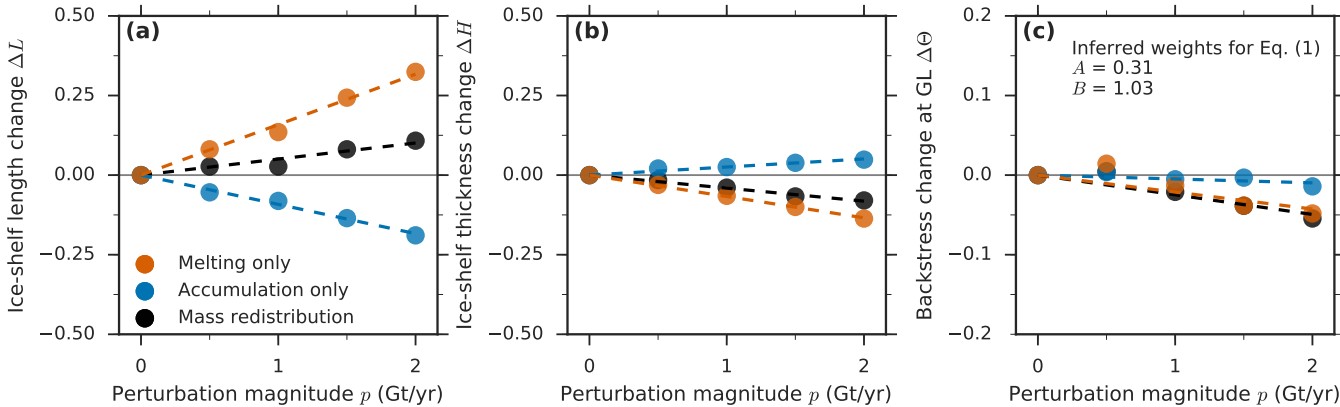

**Figure A3.** Linear regression as shown in Fig. A1 but here for the most confined lateral melt pattern ($A_{\tilde{m}} = 2 \cdot 100$ km$^2$).

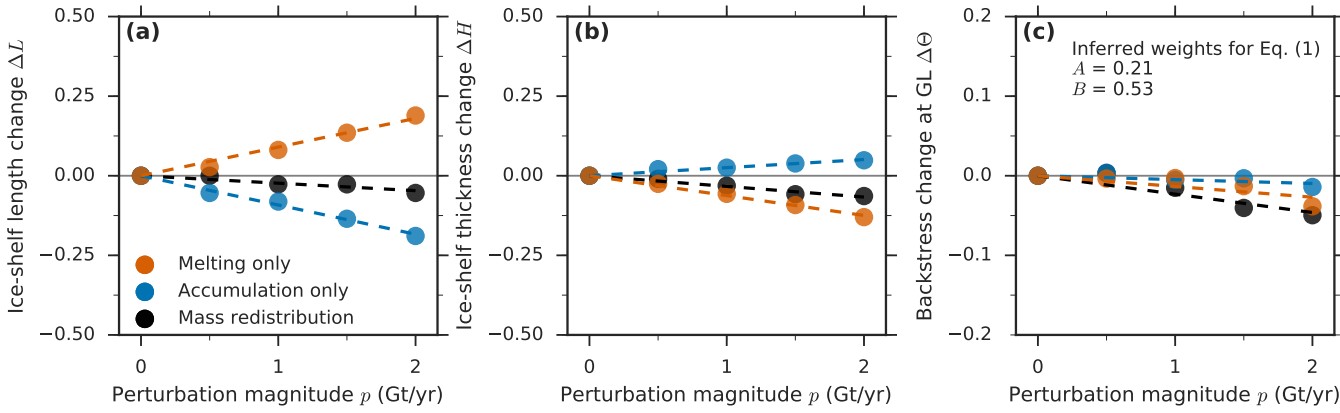

**Figure A4.** Linear regression as shown in Fig. A1 but here for the most distributed lateral melt pattern ($A_{\tilde{m}} = 2 \cdot 300$ km$^2$).





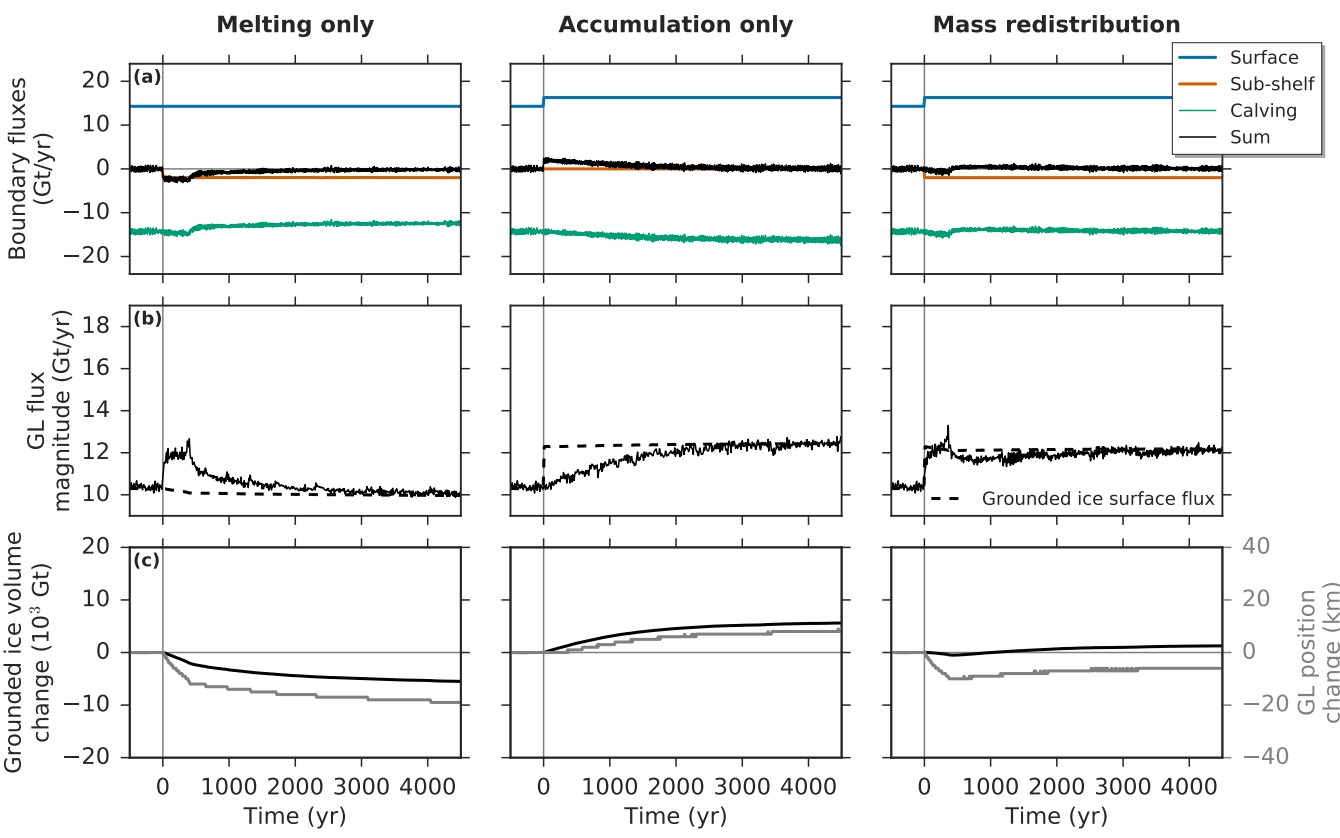

**Figure A5.** Timeseries analogous to Fig. 3 for the lateral melt perturbation. Note that due to the comparatively large lateral ice-shelf thinning resulting from this perturbation the perturbation magnitude is chosen to be only half of the magnitude applied in the central case ($\widetilde{a} = \widetilde{m} = 3\ \mathrm{Gt/yr}$ instead of $6\ \mathrm{Gt/yr}$) and the largest melt area ($A_{\widetilde{m}} = 2 \cdot 300\ \mathrm{km}^2$) is applied in order ensure that melt rates are low enough to not melt away the ice shelf locally. This smaller perturbation magnitude is reflecting in a smaller response magnitude. Qualitatively, the response curves deviate slightly for the mass-redistribution case, i.e., here the total grounding-line flux first overshoots the total surface accumulation flux before the two converge (panel b, third column) which leads to a small temporary ice-volume loss, reducing the long-term gain in ice volume (black curve, panel c).





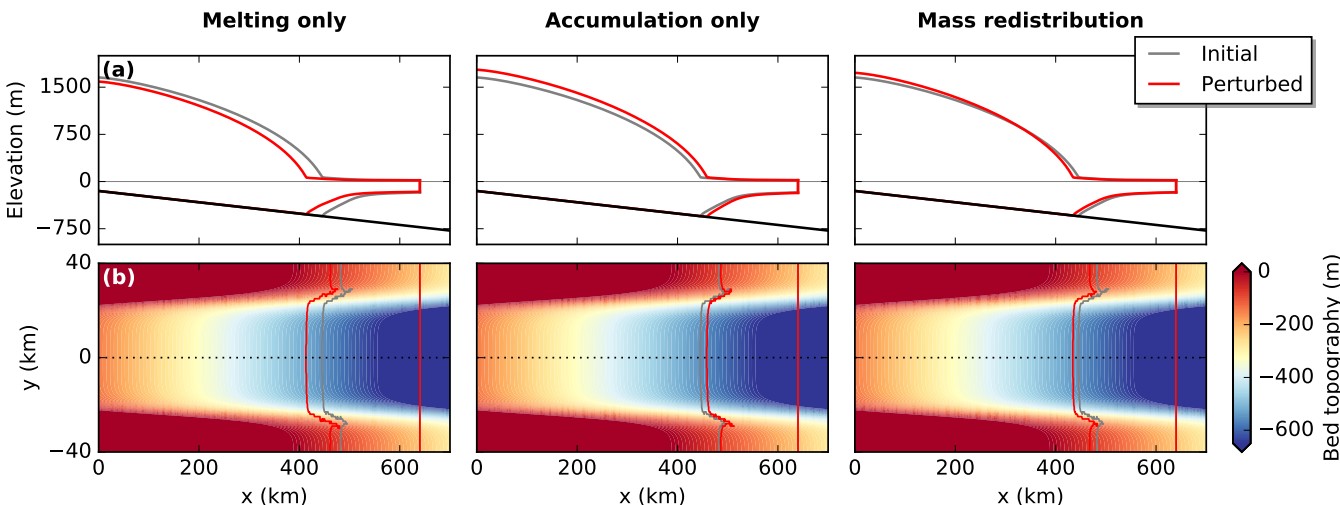

**Figure A6.** Ice-sheet profiles analogous to Fig. 4 for the lateral melt perturbation (using the same parameters as in Fig. A5). Note that despite the perturbation strength being a factor of two lower compared to the central melt example in Fig. 4, the magnitudes of grounding-line retreat are roughly the same for the "melting only" and the "mass-redistribution" forcings.



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
