# Peer review of "Snowfall versus sub-shelf melt: response of an idealized 3D ice-sheet-shelf system to mass redistribution"

_The Cryosphere, 2018_

## Referee Comment (RC1) · Anonymous Referee #1 · 23 Aug 2018

The paper entitled "Snowfall versus sub-shelf melt: response of an idealized 3D ice-sheet-shelf system to mass redistribution" by Feldmann, Reese, Winkelmann and Levermann investigates the response of a model to an increase in surface snow accumulation balanced by an increase in basal melting under its floating part. The idea is that we expect Antarctica to receive more snowfall (mass gain) and at the same time experience enhanced ocean-induced basal melting (mass loss) and it is not clear how the overall volume of the ice sheet is going to change. Here, the authors use PISM and a setup derived from MISMIP+ to investigate this question. They start the model from a steady state and apply an increase in surface accumulation over grounded ice that exactly balances the increase in basal melt applied under floating ice, and the model

runs forward until a new steady state is found. The authors find that, in all cases, the grounding line of the model does retreat, but the total volume of the system increases, leading to a net volume gain. Overall the paper is well written, easy to follow, but I have some important recommendations that should be addressed before the paper can be accepted for publication.

**1 General comments**

While I am sure that these conclusions are robust, I think that they may still be specific to this setup and may not be generalized to all systems. More specifically, I think the authors do a good job at testing the sensitivity of their conclusions to some parameters (such as ice softness or bed slopes), I think they are still missing an important parameter: basal friction.

Overall, I really struggled with is the lack of detail in the model setup. The authors do mention that they rely on the MISMIP+ experiment but do not provide any boundary conditions in the manuscript, which I think is absolutely necessary in any modeling paper. One key ingredient in the response of the model is basal friction and I could not figure out what is being used here. To be clear, I do believe that the grounding line would retreat moderately if there is a lot of basal friction, but I expect to see much more grounding line retreat for more "slippery" beds. In other words, the initial increase in basal melt is going to lead to "some" grounding line retreat, and if the ice is sliding more, the increase in ice velocity will extend far more upstream and the wave of thinning is therefore expected to propagate significantly upstream (all the way to the ice divide). The amplitude of thinning can potentially be stronger than the thickening due to a larger accumulation.

If I follow the MISMIP+ paper the authors refer to, participants are free to use any of the following laws:

- a power law

- a modified power-law relation introduced by Tsai et al. (2015)

- a modified power-law relation introduced by Schoof (2005) and explored by Gagliardini et al. (2007) and Leguy et al. (2014)

Which one is used here? I expect the model to respond differently depending on the basal friction law used, but also the friction parameter, $\beta^2$.

I recommend dividing the friction parameter $\beta^2$ by 2, 10 or even 100 to see how this impacts the model.

It is more minor but I am not 100% convinced that significantly higher snowfall has been observed in Antarctica, when significantly stronger melt has been measured, especially close to the grounding lines. So, making the argument that the anomalies will "cancel out" might not necessarily be true. On this note, I found it hard to translate the total applied perturbation, provided in Gt/yr, into accumulation/melting rates in m/yr. It would be nice to have a rough idea of what these anomalies translate into, in terms of rates. Right now, I don't know if the perturbations are realistic or not. I expect, for example, the melt to change by several 10s of m/yr, and the accumulation by a few cm/yr at most. Is that what's used here?

**2  Minor comments**

There is not a lot of punctuation in the manuscript, which makes it hard to read at times. For example, there should be a comma before "which" (most of the time), or "where". This sentence for example is clearly missing commas: "... the highest basal melt rates of Antarctic ice shelves are observed close to the grounding line where the

ice shelf is thickest which is typically ...", or "Using the model setup described above the simulations are initiated with a block ...", etc.

- p1 l2: under future warming, which leads to ... (missing comma)

- p1 l20: , which regulates (missing comma)

- p2 l9: experiments, such backstress reduction (missing comma)

- p2 l25: the pattern of sub-ice-shelf melting, we carry out (missing comma)

- p3 l1: I think the references are wrong: Morland 1987 is for SSA (you should also cite MacAyeal 1989) and Hutter 1983 is for SIA.

- p3 l5: how is melt applied on partially floating cells? A lot of recent work has shown that this might have a strong impact for resolutions on the order of 1 km or more.

- p3 l8 during ice-sheet spinput, the rate (missing comma)

- p3 l12:feeding a bay-shaped ice shelf, which (missing comma)

- p3 l26: the model described above, the simulations ... (missing comma)

- p3 l31: molten → melted

- p4 l22: This way, a set of ... (missing comma)

- p4 l25: consider replacing · by ×

- p4 l29: a bay-shaped ice shelf, which (missing comma)

- p5 l28: is characterized by grounding-line retreat *and* an increase in ice volume

- p5 l29: perturbation, here the ... (missing comma)

- p6 l2: effecting → resulting in a smaller ...

- p7 l23: In the following, we carry out (missing comma)

- p8 l15: 10% (no space between 10 and %)

- p8 l18: , which the lateral (missing comma)

- p8 l20: This fits out ... → This is consistent with ...

- p8 l25: , we show that (missing comma)

- p9 l23: reducing the concentration of sub-ice-shelf melting *close* to the grounding line?

- p10 l9: Quantification

- p10 l11: in ice-shelf length, which ... (missing comma)

- p10 l28: At the same time, out approach... (missing comma)

---

## Referee Comment (RC2) · Anonymous Referee #2 · 17 Sep 2018

In this article, the authors conducted a series of experiments to test ice-sheet response to the distribution of surface mass balance and sub-shelf melting based on the ice-sheet model PISM. Different parameters of ice properties and geometries are implemented as a sensitivity study. From the ensemble of experiments, the authors concluded that (1) the combination of increasing sub-shelf melting and surface accumulation under climate warming could result in a thicker ice sheet with smaller extent; (2) the de-buttressing effect due to lateral ice-shelf melting induces more loss on grounded ice than centeral ice-shelf melting; (3) ice-shelf thinning has more influence on upstream grounded ice than 'far-field' ice shelf removal.

**1   General comments**

The redistribution of mass perturbation result in grounding line retreat while having negative sea level contribution. I don't find this situation 'counter-intuitive' (P1L6) in the relatively stable geometry implemented, as it's a combination effect of moving ice from floating ice shelf to grounded ice and de-buttressing.

For the experimental design, the perturbation in snowfall is added inland near the ice divide, which result in higher accumulation inland than near the coast. However, in reality accumulation decreases toward inland in general. The distribution of ice mass impacts the surface slope, and thereby the driving force. The authors mentioned an initial examination of changing perturbation zones has been conducted (P4L1). I suggest showing the difference in the supplementary.

The rather narrow range of parameters make the results limited and so less compelling. For example, instead of using the retrograde slope as Asay-Davis et

al. (2016), the bed slopes range is 1.0, 1.1 and 1.2. However, the bed slopes of Antarctica basins can be flatter and negative. Will the retreat-mass-gain conclusion fail in that case? Similarly, the width difference between experiments is 8 km at most, while in reality the width/length ratio range is much larger and can be >1. The ice sheet response to different perturbations and the coefficients of the backstress formula could be different. The choice of parameters exclude the West Antarctic geometry and wider ice shelves, therefore more experiments are needed to confirm the conclusions.

**2   Minor comments**

P2L10: or $\rightarrow$ and

P4L25: 9*2*3 $\rightarrow$ 12*2*3? There are four parameters A,s,a,$w_c$ with 3 values for each.

P6L3: extra quotation mark

P6L19: maybe add a figure of velocity to show the 'low advection of ice from the lateral ridges'

P7L5: maybe add a figure of the ice shelf cross section to compare the ice-shelf thinning after years of different perturbations

P7L31: Formulas or references needed to explain how you calculate the back-stress

P21L20: a bit confusing. The grounding-line retreat is stronger at the sides than in the center, but the grounding-line is retreating overall. Why does $\Delta L$ decrease?

---

## Author Comment (AC1) · 15 Oct 2018

**Detailed response to the Editor on manuscript tc-2018-109**

"Snowfall versus sub-shelf melt: response of an idealized 3D ice-sheet-shelf system to mass redistribution"

by J. Feldmann, R. Reese, R. Winkelmann and A. Levermann

Dear Editor Olaf Eisen,

We would like to thank you for handling the review process and the reviewers for their detailed look at our manuscript. We are happy for the positive assessment of the Reviewer #1 and would like to thank both reviewers for their valuable and constructive comments and suggestions, which were very helpful and improved our manuscript.

Following the reviewers' main requests, i.e., to conduct more experiments in order to support our findings, we carried out three different sets of additional simulations. Based on the first set of new simulations, we analyze the role of basal friction in our model setup, which was the main concern by Reviewer #1 (see new Appendix C "Sensitivity to changes in basal friction" with Fig. C1). Second, we investigate the influence of the location of the surface-accumulation perturbation, as suggested by Reviewer #2 (see new Appendix B "Sensitivity to changes in accumulation area" with Figs. B1, B2). Third, we prescribe a landward down-sloping bed topography instead of the previous landward up-sloping bed, addressing the other major concern of Reviewer #2 regarding the variety of model parameters, in particular the bed topography (see new Appendix D "Role of bed-slope direction" with Figs. D1-3). To this end, we also revised the discussion section of the manuscript, emphasizing the limitations of our experiments. Picking up the ideas of Reviewer #2, we added two more figures to the manuscript (Figs. 8 and C2). We also included a visualization of the ice-shelf buttressing field diagnosed from our simulations (Fig. A1). Due to the abundance of figures in our manuscript we shifted five of them from the Appendix to the Supplement (original Figs. A2-A6 now named Figs. S1-S5).

Please find below the *Reviewer's comments in italics* and our detailed response in blue. We have further attached a revised manuscript that highlights the changes in the submission, as well as a clean revised version.

Best wishes,
Johannes Feldmann on behalf of all the co-authors
*The paper entitled "Snowfall versus sub-shelf melt: response of an idealized 3D ice-sheet-shelf system to mass redistribution" by Feldmann, Reese, Winkelmann and Levermann investigates the response of a model to an increase in surface snow accumulation balanced by an increase in basal melting under its floating part. The idea is that we expect Antarctica to receive more snowfall (mass gain) and at the same time experience enhanced ocean-induced basal melting (mass loss) and it is not clear how the overall volume of the ice sheet is going to change. Here, the authors use PISM and a setup derived from MISMIP+ to investigate this question. They start the model from a steady state and apply an increase in surface accumulation over grounded ice that exactly balances the increase in basal melt applied under floating ice, and the model runs forward until a new steady state is found. The authors find that, in all cases, the grounding line of the model does retreat, but the total volume of the system increases, leading to a net volume gain. Overall the paper is well written, easy to follow, but I have some important recommendations that should be addressed before the paper can be accepted for publication.*

We would like to thank the Referee for taking the responsibility to carefully review our manuscript. We are glad to receive a positive assessment of our manuscript and are grateful for the Reviewer's constructive comments and suggestions.

*1 General comments*
*While I am sure that these conclusions are robust, I think that they may still be specific to this setup and may not be generalized to all systems. More specifically, I think the authors do a good job at testing the sensitivity of their conclusions to some parameters (such as ice softness or bed slopes), I think they are still missing an important parameter: basal friction. Overall, I really struggled with is the lack of detail in the model setup. The authors do mention that they rely on the MISMIP+ experiment but do not provide any boundary conditions in the manuscript, which I think is absolutely necessary in any modeling paper.*

We completely agree with the Reviewer that a numerical modeling study, like the one presented here, needs a detailed description of the model setup. Our model setup is largely based on the MISMIP+ setup for which a model description paper is available open source (Asay-Davis et al., 2016). To make this clearer, we modified the "Model setup" section of the manuscript (P3L8-11) and to keep the section concise and avoid redundancy, we refer the interested reader to this description paper for details. Regarding the lacking detail on the boundary conditions used in our

experiments mentioned by the Reviewer we amended our manuscript (P3L4-5, P3L10), in particular with respect to the basal-friction law (P3L13-15, see Reviewer comment and our answer below).

*One key ingredient in the response of the model is basal friction and I could not figure out what is being used here. To be clear, I do believe that the grounding line would retreat moderately if there is a lot of basal friction, but I expect to see much more grounding line retreat for more "slippery" beds. In other words, the initial increase in basal melt is going to lead to "some" grounding line retreat, and if the ice is sliding more, the increase in ice velocity will extend far more upstream and the wave of thinning is therefore expected to propagate significantly upstream (all the way to the ice divide). The amplitude of thinning can potentially be stronger than the thickening due to a larger accumulation.*

*If I follow the MISMIP+ paper the authors refer to, participants are free to use any of the following laws:*

*• a power law*

*• a modified power-law relation introduced by Tsai et al. (2015)*

*• a modified power-law relation introduced by Schoof (2005) and explored by Gagliardini et al. (2007) and Leguy et al. (2014)*

*Which one is used here? I expect the model to respond differently depending on the basal friction law used, but also the friction parameter, $\beta^2$.*

Thanks for pointing out this important detail. Indeed, we did not clarify that we use the unmodified power law in our study. Now we clearly state in the Methods section the type of friction law we chose for our simulations (P3L13-15) and also give the formula in the Appendix C (Eq. C1). We also tested how more "slippery" beds affect our result (see response below).

*I recommend dividing the friction parameter $\beta^2$ by 2, 10 or even 100 to see how this impacts the model.*

As mentioned by the Reviewer, the basal friction parameter $\beta^2$ ($\beta^2=C$ in our Eq. A6) is relevant for ice-sheet dynamics and indeed worth incorporating into our study. Following the Reviewer's suggestion we divided the parameter by 2, 10 and 100, though dividing by 100 did not yield convergence of the SSA; in this case the bed might simply be too slippery. We discuss the results for *C/2* and *C/10* in a new section (Appendix C) and added a new corresponding figure (Fig. C1). Also, the we updated Figs. 5-8 with the new data (grounding-line change, volume change, flux change, buttressing sensitivity) and inserted the parameter values into Table 1. We found that changing the basal slipperiness did not change our results qualitatively. However, as the Reviewer suggests, the quantitative response is reduced (see Fig. C1) which we now also mention in the discussion (P9L31-33).

*It is more minor but I am not 100% convinced that significantly higher snowfall has been observed in Antarctica, when significantly stronger melt has been measured, especially close to the grounding lines.  So, making the argument that the anomalies will "cancel out" might not necessarily be true. On this note, I found it hard to translate the total applied perturbation, provided in Gt/yr, into accumulation/melting rates in m/yr. It would be nice to have a rough idea of what these anomalies translate into, in terms of rates. Right now, I don't know if the perturbations are realistic or not.  I expect, for example, the melt to change by several 10s of m/yr, and the accumulation by a few cm/yr at most. Is that what's used here?*

To avoid misunderstandings, we would like to point out that in our manuscript we neither want to claim that a simultaneous increase in sub-shelf melting and snowfall has been or will be observed locally nor that both would have to cancel out. A general increase in both forcings that might be expected in the future is the motivation for our conceptual, simplified experiments, for which we make the very simple assumption that the two forcings are equivalent in terms of mass addition/reduction. To be more clear on that, we added a corresponding sentence to the discussion (P12L14-16). Regarding the translation of the accumulation rates into m/yr mentioned by the Reviewer, we included the numbers into the manuscript (for melting rates  in m/yr see P12L13), also setting them into context by a comparison to observational data (P12L4-9).

**2  Minor comments**
*There is not a lot of punctuation in the manuscript,  which makes it hard to read at times.  For example, there should be a comma before "which" (most of the time), or "where".  This sentence for example is clearly missing commas:  "…  the highest basal melt rates of Antarctic ice shelves are observed close to the grounding line where the ice shelf is thickest which is typically …", or "Using the model setup described above the simulations are initiated with a block …", etc*

We would like to thank the Reviewer for this detailed look onto the punctuation, pointing to the spots where we missed placing a comma. We inserted these, which indeed improves readability.

- *p1 l2: under future warming, which leads to … (missing comma)*
Done.
- *p1 l20: , which regulates (missing comma)*
Done.
- *p2 l9: experiments, such backstress reduction (missing comma)*
Done.
- *p2 l25: the pattern of sub-ice-shelf melting, we carry out (missing comma)*
Done.
- *p3 l1: I think the references are wrong: Morland 1987 is for SSA (you should also cite MacAyeal 1989) and Hutter 1983 is for SIA.*

Thanks for the hint. We indeed mixed up the references for SIA and SSA here. We also added the reference suggested by the Reviewer.

- *p3 l5: how is melt applied on partially floating cells? A lot of recent work has shown that this might have a strong impact for resolutions on the order of 1 km or more.*

This is truly an important information for the reader that we were missing and we are glad that the Reviewer points to it. In fact, in contrast to basal friction we do not interpolate basal melt rates and thus do not apply melting to partially floating cells. We added this statement to the manuscript (P3L7) as suggested by the Reviewer.

- *p3 l8 during ice-sheet spinput, the rate (missing comma)*

Done.

- *p3 l12:feeding a bay-shaped ice shelf, which (missing comma)*

Done.

- *p3 l26: the model described above, the simulations ... (missing comma)*

Done.

- *p3 l31: molten→melted*

Done.

- *p4 l22: This way, a set of ... (missing comma)*

Done.

- *p4 l25: consider replacing · by ×*

Done. We would like to leave the multiplication sign as it is.

- *p4 l29: a bay-shaped ice shelf, which (missing comma)*

Done.

- *p5 l28: is characterized by grounding-line retreat and an increase in ice volume*

Done.

- *p5 l29: perturbation, here the ... (missing comma)*

Done.

- *p6 l2: effecting → resulting in a smaller ...*

Done.

- *p7 l23: In the following, we carry out (missing comma)*

Done.

- *p8 l15: 10% (no space between 10 and %)*

Done.

- *p8 l18: , which the lateral (missing comma)*

Done.

- *p8 l20: This fits out ... → This is consistent with ...*

Done.

- *p8 l25: , we show that (missing comma)*

Done.

- *p9 l23: reducing the concentration of sub-ice-shelf melting close to the grounding line?*

Done.

- *p10 l9: Quantification*

Done.

- *p10 l11: in ice-shelf length, which ... (missing comma)*

Done.

- *p10 l28: At the same time, out approach... (missing comma)*

Done.
*In this article, the authors conducted a series of experiments to test ice-sheet response to the distribution of surface mass balance and sub-shelf melting based on the ice-sheet model PISM. Different parameters of ice properties and geometries are implemented as a sensitivity study. From the ensemble of experiments, the authors concluded that (1) the combination of increasing sub-shelf melting and surface accumulation under climate warming could result in a thicker ice sheet with smaller extent; (2) the de-buttressing effect due to lateral ice-shelf melting induces more loss on grounded ice than centeral ice-shelf melting; (3) ice-shelf thinning has more influence on upstream grounded ice than 'far-field' ice shelf removal.*

We would like to thank the reviewer for taking time and effort to review our manuscript. We are thankful for the constructive criticism, the ideas and suggestions of the Reviewer.

*1 General comments*
*The redistribution of mass perturbation result in grounding line retreat while having negative sea level contribution. I don't find this situation 'counter-intuitive' (P1L6) in the relatively stable geometry implemented, as it's a combination effect of moving ice from floating ice shelf to grounded ice and de-buttressing.*

We agree with the Reviewer that the phrase counter-intuitive might be vague and irritating in this context and thus removed it.

*For the experimental design, the perturbation in snowfall is added inland near the ice divide, which result in higher accumulation inland than near the coast. However, in reality accumulation decreases toward inland in general. The distribution of ice mass impacts the surface slope, and thereby the driving force. The authors mentioned an initial examination of changing perturbation zones has been conducted (P4L1). I suggest showing the difference in the supplementary.*

Following the suggestion of the Reviewer, we included the analysis into the new Appendix B, briefly discussing the influence of a change in the snowfall perturbation zone and visualizing the results in new Fig. B2. According to the Reviewer's remarks, the investigation includes the shifting of the snowfall concentration towards the coast (see perturbation zones in new Fig. B1). We overall found our results to be robust according to the placement of the snowfall perturbation with slight quantitative differences (discussed in Appendix B and in the discussion section [P9L31-33]).

*The rather narrow range of parameters make the results limited and so less compelling. For example, instead of using the retrograde slope as Asay-Davis et al. (2016), the bed slopes range is 1.0, 1.1 and 1.2. However, the bed slopes of Antarctica basins can be flatter and negative. Will the retreat-mass-gain conclusion fail in that case?*

We understand the Reviewer's concern regarding the shape of the used bed topography, as it limits our results to the case of a landward up-sloping bed. Note that the magnitude of the prescribed bed slope is actually not on the order of 1 but substantially flatter (on the order of 10^-2). This bed slope is increased by a scaling factor *s* of 1.1 and 1.2, respectively (see P3L13-14 for the formula), making a difference in grounding-line position of about 100 km. Now we refer to this range (P11L4-5) and the limitations due to parameter choices (P12L18-20) remarked by the Reviewer. For comparison, the used bed slope is one order of magnitude steeper than, e.g., the landward up-sloping bed of Bindschadler Ice Stream in West Antarctica (Ross et al., Nat. Geosc., 2012).

The original idea behind using a linearly landward up-sloping bed was to ease the analysis of the ice-sheet behavior. For instance, in this case one does not have to care about instability or about a possible switch in the sign of the bed slope in the course of grounding line migration due to the applied forcing or parameter variation. Nevertheless, we carried out additional simulations prescribing the retrograde bed topography as suggested by the Reviewer. The results are summarized in Appendix D with new Figs. D1-D3. Overall the new results are in line with our previous ensemble analysis, in particular confirming the retreat-mass-gain response mentioned by the Reviewer, which we briefly discuss in the new section. We amended the discussion section accordingly, attributing the similarity between the results to the strong influence of buttressing in our simulations (P10L8-13, P11L23).

*Similarly, the width difference between experiments is 8 km at most, while in reality the width/length ratio range is much larger and can be > 1. The ice sheet response to different perturbations and the coefficients of the backstress formula could be different. The choice of parameters exclude the West Antarctic geometry and wider ice shelves, therefore more experiments are needed to confirm the conclusions.*

We agree with the Reviewer that the difference in the channel width is limited throughout our simulations. The maximum width of $2 \ast w_c$ = 64 km we use here is determined by the y-dimension of the setup. For a larger width no proper ice-shelf confinement emerges in the simulations making them unsuitable for this study. Beyond the minimum width of 48 km the ice shelf becomes too short to allow for a proper application of central/lateral melt perturbation of different areal extent, which is our purpose in this paper. The Reviewer mentions a maximum channel-width difference of 8 km, which is indeed the maximum difference in the parameter $w_c$. In fact, $w_c$ is the half width of the bed channel which we define in Table 1 but accidentally called "channel width" in the Methods section, for which we want to apologize (now corrected, P4L25). Consequently, the effective maximum channel-width difference throughout our experiments is 16 km, making a change of 1/3 of the default width of the channel. Calculating the width/length ratios of the simulated steady-state ice shelf confinements, mentioned by the Reviewer, we find that this ratio $R$ varies between 0.66 and 1.72. Thus our ensemble of simulations account for both $R<1$ and $R>1$. However, we are aware that due to the limitations in our model setup the conducted

simulations only cover a small part of possible shapes of ice-sheet-shelf systems/ice-sheet outlets. We revised the discussion section accordingly (P11L26-29, P12L18-22), also weakening our statements on the applicability of our results to the real world (P11L26-29). In order to account for a larger variety of simulated ice-sheet-shelf systems the model domain would have to be re-designed, e.g., widened, in turn requiring a tuning of other parameters (ice softness, accumulation, basal friction). Though such an examination might of course be very interesting, we think that it is beyond the scope of this study and the computational resources available to us.

**2  Minor comments**

P2L10:  or → and
Done.

P4L25:  9*2*3 → 12*2*3?  There are four parameters A,s,a,w_c with 3 values for each.

Indeed, we have these 4 parameters with 3 values for each parameter. Due to the fact that one value of each parameter is already reserved for the default set of parameters, for each parameter there remain two values, creating 4*2=8 further parameter sets.  In order to be more clear on that we added a phrase to the manuscript. Since we included basal friction as a fifth parameter into our analysis in the course of the review process, now the ensemble size is 11*2*3 = 66 (P4L28, P4L34).

P6L3:  extra quotation mark
Done.

P6L19: maybe add a figure of velocity to show the 'low advection of ice from the lateral ridges'
This is a good idea to show the velocity field to support our argument. Adding a corresponding figure (Fig.  C2), we also show the velocity field for reduced basal friction in panel b.

P7L5:  maybe add a figure of the ice shelf cross section to compare the ice-shelf thinning after years of different perturbations
Following the Reviewer's suggestion, we added a new figure to the manuscript (new Fig. 8) exemplary showing the the different thinning patterns after 10 yr and 40 yr, respectively, corresponding to the results in Fig. 7.

P7L31: Formulas or references needed to explain how you calculate the back-stress
We thank the Reviewer for pointing out this important detail. We inserted a reference to De Rydt et al, 2015, Eq. (1), where the formula for the backstress calculation is given (P21L11-12). We also included a figure showing the calculated steady-state ice-shelf backstress field (Fig A1).

P21L20:  a bit confusing.  The grounding-line retreat is stronger at the sides than in the center, but the grounding-line is retreating overall.  Why does $\Delta L$ decrease?

Our intention with this sentence is to simply state that grounding line retreat is stronger at the margins than in the center, causing the decrease in ice-shelf length. We modified the sentence accordingly (P22L20-21).